# Validating childhood asthma in an epidemiological study using linked electronic patient records

Rosaleen P Cornish,[1] John Henderson,[1] Andrew W Boyd,[1] Raquel Granell,[1] Tjeerd Van Staa,[2] John Macleod[1]

▶ Prepublication history and additional material is available. To view please visit the journal (http://dx.doi.org/10.1136/bmjopen-2014-005345).

[1]School of Social and Community Medicine, University of Bristol, Bristol, UK
[2]Clinical Practice Research Datalink (CPRD), Medicines and Healthcare products Regulatory Agency, London, UK

**Correspondence to**
Dr John Macleod;
john.macleod@bristol.ac.uk

## ABSTRACT

**Objective:** To investigate the performance of parent-reported data in identifying physician-confirmed asthma.

**Design and setting:** Validation study using linkage between the Avon Longitudinal Study of Parents and Children (ALSPAC) and electronic patient records held within the General Practice Research Database (GPRD).

**Participants:** Participants were those eligible to participate in ALSPAC who also had a record in the GPRD; this included 765 individuals, just under 4% of ALSPAC-eligible participants. The analysis was based on 141 participants with complete parent-reported asthma data.

**Primary and secondary outcome measures:** The main GPRD outcome measure was whether a child had a diagnosis of asthma before they were nine. Parent-reported measures were doctor diagnosis of asthma (before mean age 7.5 years), various outcomes based on wheezing and breathlessness recorded longitudinally between 6 months and 8.5 years. Secondary outcomes were bronchial hyper-responsiveness (BHR), forced expiratory volume in 1 s/forced vital capacity ratio and skin prick test responses.

**Results:** Among the 141 participants with complete parent-reported data, 26 (18%) had an asthma diagnosis before age nine. Using general practitioner (GP)-recorded asthma as the gold standard, the question 'Has a doctor ever diagnosed your child with asthma?' was both sensitive (88.5%) and specific (95.7%). 'Ever wheezed' had the highest sensitivity (100%) but low specificity (60%). More specific definitions were obtained by restricting to those who had wheezed on more than one occasion, experienced frequent wheeze and/or wheezed after the age of 3, but these measures had low sensitivities. BHR only identified 50% of those with a GP-recorded diagnosis.

**Conclusions:** Parental reports of a doctor's diagnosis agree well with a GP-recorded diagnosis. High specificity for asthma can be achieved by using detailed wheezing questions, although these definitions are likely to exclude mild cases of asthma. Our study shows that linkage between observational studies and electronic patient records has the potential to enhance epidemiological research.

## Strengths and limitations of this study

- We were able to successfully link data from the Avon Longitudinal Study of Parents and Children (ALSPAC) with the General Practice Research Database (GPRD).
- The richness of the ALSPAC data allowed us to explore a number of different epidemiological constructs of asthma definition in relation to a recorded physician diagnosis.
- We were able to compare the relative performance of objective measurements and lung function to that of parent-reported symptom data in identifying physician-confirmed asthma.
- Information on wheezing was collected longitudinally between ages 6 months and 8.5 years.
- The proportion of participants with complete self-reported data from ALSPAC was relatively small.

## INTRODUCTION

Asthma is difficult to measure in epidemiological studies because there are no definitive diagnostic criteria. The presence of recurrent respiratory symptoms, particularly wheezing and breathlessness, is central to its diagnosis in children. Bronchial hyper-responsiveness (BHR) is also a feature of asthma[1 2] but is not consistently present and can exist in the absence of clinical symptoms. Other measures of lung function may be normal during asymptomatic periods and can also be abnormal in the presence of other respiratory diseases.[1] Thus, owing to the variable nature of asthma symptoms—both between and within individuals—its diagnosis remains a clinical one and clinical assessment is regarded as the best method for validating self-reported asthma data.[3]

However, clinical examination is costly and questionnaires are often preferred in epidemiological studies. Questions on wheezing, such as those used in the International Study of Asthma and Allergies in Childhood (ISAAC),[4] have been shown to provide valid

measures of the prevalence of asthma in children.[5–8] Having said this, it is not possible to reliably distinguish between wheezing due to viral respiratory infections, which are common among young children, and wheezing due to asthma. In addition, mistaking other respiratory noises as wheezing can lead to over-reporting.[7 9 10] Consequently, it has been suggested that questionnaires should be supplemented by measurements of airway responsiveness.[11] However, several validation studies have shown that agreement between such measures and clinical asthma is poor.[3]

In the UK, childhood asthma is predominantly diagnosed and managed in primary care. Linkage to primary care records therefore provides a potential means to validate self-reported data. In this study, we have compared longitudinal data on wheezing and other respiratory outcomes with linked electronic patient records to investigate the performance of epidemiological data in identifying physician-confirmed asthma.

## METHODS
### Subjects
Participants were those eligible to participate in the Avon Longitudinal Study of Parents and Children (ALSPAC) who also had a record in the General Practice Research Database (GPRD; now the Clinical Practice Research Datalink: http://www.cprd.com). The GPRD is an anonymised database of primary care records of around five million patients in the UK. ALSPAC has been described in detail previously.[12] Briefly, just over 20 000 pregnant women living in and around Bristol, UK with due dates during 1991–1992 were eligible to take part; 15 247 pregnancies were recruited (15 390 fetuses, of which 14 701 were alive at 1 year) and these have been followed up since birth (the ALSPAC website has a searchable data dictionary; http://www.bris.ac.uk/alspac/researchers/data-access/data-dictionary/ describing all available data).

### Linkage between ALSPAC and the GPRD
Linkage between ALSPAC and the GPRD was conducted by the National Health Service Information Centre (NHS IC) as a trusted third party, using a methodology to preserve anonymity. They had previously linked ALSPAC participants to the NHS Central Register, with a 99% match rate[12]; this was carried out on the basis of NHS ID number, name, date of birth and postcode using deterministic linkage. With approval from the NHS National Information Governance Board (NIGB) Ethics and Confidentiality Committee, the NHS IC used this information to identify ALSPAC eligible individuals who also appeared in the GPRD; they then sent an anonymised linking dataset to be stored securely at the GPRD. ALSPAC and GPRD data for linked individuals were merged and analysed in a safe setting at the GPRD offices. As GPRD is anonymous and collected on an opt-out basis, and anonymity was preserved using the safeguards described above, this piece of research does not require consent above and beyond the consent obtained for participation in ALSPAC. However, ALSPAC has recently been collecting consent from participants, who are now adults, for ongoing participation in the study as well as consent to extract information from health and other administrative records, and any participants who withdrew from the study or did not agree to their health records being extracted were excluded from the linkage.

### ALSPAC data
When the children were, on average, 7.5 years old, mothers were asked (via a postal questionnaire) whether a doctor had ever diagnosed their child with asthma. Data on asthma symptoms (wheezing and breathlessness) were collected at 6 months and approximately every 12 months thereafter (at 18, 30, 42, 57, 69, 81, 91 and 103 months). As well as asking whether the child had wheezed, the mothers were asked whether their child had experienced 'wheezing with whistling on the chest when (s)he breathed' and, if so, on how many occasions. Symptom questions referred to the previous 12 months, apart from at 6 and 57 months, which referred to the past 6 and 15 months, respectively. A child was defined to have wheezed if the response to either question about wheezing was yes and not wheezed if the response to both was no. Frequent wheeze was defined as wheezing with whistling on the chest on at least three occasions during the past 12 months. During a clinic attended between 8 and 9 years, forced expiratory volume in 1 s ($FEV_1$) and forced vital capacity (FVC) were measured by spirometry and bronchial responsiveness was measured using the rapid methacholine inhalation test[13]; the dose–response slope of $FEV_1$ per μmol methacholine was used to classify participants into four categories: none, low, moderate and high, based on tertiles of the dose–response slope; BHR was defined as being the highest tertile. Sensitisation to house dust mite, cat and mixed grass was assessed by skin prick tests with a positive response defined as a mean weal diameter of 2 mm or greater; this definition identified over 90% of participants sensitised to any one of a panel of up to 12 allergens.[14] Symptom data were converted to 'ever wheezed', 'ever reported frequent wheeze' and 'ever had breathlessness'—all defined as positive if there was a positive response at any time point and negative if all responses were negative. Participants were classified according to whether or not their $FEV_1$/FVC ratio was less than 85% and whether or not they had at least one positive skin prick test.

### GPRD data
Patients were defined as having a general practitioner (GP) diagnosis of asthma if they had a Read code (the clinical coding system used by GPs in the UK—http://

www.connectingforhealth.nhs.uk/systemsandservices/data/uktc/readcodes/index_html) indicating an asthma diagnosis (any Read code starting with 'H33') in their record. In order to coincide with the timing of ALSPAC measurements, participants were classified according to whether they had a diagnosis before 9 years of age. Thus, participants who had no asthma diagnosis but left a GPRD practice before nine were excluded as it could not be determined whether they were subsequently diagnosed with asthma. Conversely, those who were not in the GPRD from birth were not excluded because historic diagnoses appeared in patients' records.

## Statistical methods

Sensitivity, specificity and predictive values for ALSPAC outcomes and combinations of outcomes were calculated using GP-recorded diagnosis as the gold standard. Exact CIs were calculated based on binomial probabilities. These analyses were restricted to those with complete parent-reported data from ALSPAC. Because the question about a doctor's diagnosis of asthma was asked, on average, a year earlier than the last wheezing question as well as before lung function was measured, this outcome was also compared to GP-recorded asthma before age 8 years in order to evaluate the likely impact of this age gap on the estimated sensitivity and specificity. In order to try to restrict to current asthma, we looked at whether children with an early diagnosis had evidence of persistent asthma by school age or beyond. There was only one child in the complete-case analysis with an early diagnosis (aged 3) but no further evidence of asthma. However, since this child left the GPRD at age 6, it could not be determined whether or not they had persistent asthma and they were left in the analysis. Analyses were carried out using Stata V.12.0.

## RESULTS

Of the live births linked by the NHS IC, 765 appeared in the GPRD (this constitutes 4% of those eligible to take part in ALSPAC); 4 individuals did not want their health records accessed and 1 registered into and left the GPRD on the same day. Of the remaining 760 individuals, 61 without an asthma diagnosis transferred out of the GPRD before the age of 9, leaving 699 participants with known asthma status. Of these, 488 (70%) had enrolled in ALSPAC, 251 (51%) had information on whether a doctor had ever diagnosed asthma, 141 (29%) had complete parent-reported asthma data; slightly fewer had BHR, $FEV_1$, FVC and skin prick test data. Characteristics of all linked individuals with known asthma status and those included in the analysis are shown in table 1. Among the former, there were 115 children with a diagnosis of asthma before age 9, giving a cumulative incidence of 16%. This figure was slightly higher (18%) among those with complete ALSPAC data; these children were also less likely to live in deprived areas, as indicated by the index of multiple deprivation (IMD 2007; $\chi^2$=18.8, p=0.001).

Table 2 shows the proportions with and without a GP diagnosis correctly identified by different ALSPAC variables, as well as the overall percentage reporting each outcome. Wheezing was common: 51% of the children had ever wheezed and 27% reported frequent wheeze on at least one occasion.

Of the ALSPAC variables, 'ever wheezed' had the highest sensitivity (100%; 95% CI 86.8% to 100%) but low specificity (60%; 50.4% to 69.0%). The question 'Has a doctor ever diagnosed your child with asthma?' was both sensitive (88.5%; 69.8% to 97.6%) and specific (95.7%; 90.1% to 98.6%). By restricting to those who had wheezed on more than one occasion, experienced frequent wheeze or wheezed after the age of 3, more specific asthma definitions were obtained (table 2 and see online supplementary table S1). However, the consequent losses in sensitivity were substantial. Positive and negative predictive values are given in table 3. These confirm the above results. BHR only identified 50% of those with a GP diagnosis; the specificity was also relatively low (75%). The sensitivities of $FEV_1$/FVC ratio and skin prick tests were also low (see online supplementary table S2).

Combining symptoms with reports of a doctor's diagnosis of asthma gave slightly higher specificities than the latter alone (table 4), but the sensitivity of each

**Table 1** Characteristics of all linked participants with known asthma status compared to those with complete ALSPAC parent-reported data

| GPRD-recorded characteristics | Linked participants who did not exit GPRD before age 9 (n=699) | Participants with complete ALSPAC parent-reported data* (n=141) |
|---|---|---|
| Sex—male | 348 (50%) | 70 (50%) |
| $IMD^2$ quintile—least deprived | 68 (13%) | 24 (24%) |
| Second | 155 (30%) | 30 (30%) |
| Third | 120 (24%) | 27 (27%) |
| Fourth | 58 (11%) | 8 (8%) |
| Most deprived | 109 (21%) | 11 (11%) |
| Asthma—cumulative incidence | 115 (16%) | 26 (18%) |

*The denominator for IMD is slightly lower (510 for all linked participants with GPRD asthma status at age 9; 100 for those with complete ALSPAC parent-reported data) as this variable is not complete within GPRD.
ALSPAC, Avon Longitudinal Study of Parents and Children; GPRD, General Practice Research Database; IMD, index of multiple deprivation.

**Table 2** Cross tabulation of key ALSPAC asthma variables and GPRD asthma

| ALSPAC asthma outcome | GP-recorded diagnosis of asthma | | |
|---|---|---|---|
| | No | Yes | Overall |
| Doctor diagnosis of asthma by 7.5 years | | | |
| No | 110 (95.7%) | 3 | 113 |
| Yes | 5 | 23 (88.5%) | 28 (19.9%) |
| Ever reported breathlessness in the past 12 months | | | |
| No | 96 (83.5%) | 6 | 102 |
| Yes | 19 | 20 (76.9%) | 39 (27.7%) |
| Ever reported wheezing in the past 12 months | | | |
| No | 69 (60.0%) | 0 | 69 |
| Yes | 46 | 26 (100%) | 72 (51.1%) |
| Wheezing reported on at least two occasions | | | |
| No | 88 (76.5%) | 3 | 91 |
| Yes | 27 | 23 (88.5%) | 50 (35.5%) |
| Wheezing reported after the age of 3 years | | | |
| No | 96 (83.5%) | 9 | 105 |
| Yes | 19 | 17 (65.4%) | 36 (25.5%) |
| Ever reported a frequent wheeze | | | |
| No | 96 (83.5%) | 7 | 103 |
| Yes | 19 | 19 (73.1%) | 38 (27.0%) |
| Reported a frequent wheeze after the age of 3 years | | | |
| No | 110 (95.7%) | 11 | 121 |
| Yes | 5 | 15 (57.7%) | 20 (14.2%) |

ALSPAC, Avon Longitudinal Study of Parents and Children; GP, general practitioner; GPRD, General Practice Research Database.

combination was somewhat lower. The combination with the highest Youden's index (sensitivity+specificity−100%) was wheezing reported on at least two occasions plus reports of a doctor's diagnosis (Youden's index 74.3%, compared with a figure of 84.2% for doctor's diagnosis alone). This combination also gave high predictive values, as did combining frequent wheeze with a doctor's diagnosis.

Changing the age cut-off in GPRD from 9 to 8 years for the question 'Has a doctor ever diagnosed your child with asthma?' had only a small impact on the results: the sensitivity of this measure increased to 91.3% and the specificity decreased to 94.1%.

## DISCUSSION

We have found that parental reports of a doctor's diagnosis accurately predicted the cumulative incidence of asthma in children with complete data recorded on asthma symptoms from birth to (8.5) years. Definitions consisting of wheeze or frequent wheeze reported on several occasions were more specific but had low sensitivity. This is unsurprising, as these criteria would favour more severe cases. Conversely, the sensitivity of ever wheezing and wheezing reported on two or more occasions was high but both had comparatively low specificities.

An issue in asthma validation studies is that there is no true gold standard. However, clinical assessment is regarded as the optimal standard.[3] In this study, we have used linkage to the GPRD to identify ALSPAC participants with an asthma diagnosis in their GP record. Although diagnoses in the GPRD have been shown to be well recorded, particularly for chronic conditions,[15] there are some weaknesses with this approach. Low population coverage of the GPRD meant that only a small proportion of ALSPAC participants were captured through this linkage. Further, linked individuals only had a complete record while registered with a practice contributing to the GPRD. Having said this, of 165 participants in this study who ever had an asthma diagnosis, 65 (39%) were dated before their GPRD registration date. Also, high agreement between parent reports and GP records suggest that missing historic diagnoses were not a substantial issue. A further shortcoming is the extent of missing ALSPAC data: complete data were only available for 29% of the linked participants with known asthma status. If those without complete data were less likely to complete the questionnaires accurately than those with complete data, then the levels of agreement

**Table 3** PPV and NPV of different asthma outcome measures

| ALSPAC asthma outcome | PPV (95% CI) | NPV (95% CI) |
|---|---|---|
| Single outcomes | | |
| Doctor diagnosis of asthma | 82.1% (63.1% to 93.9%) | 97.3% (92.4% to 99.4%) |
| Ever reported breathlessness | 51.3% (34.8% to 67.6%) | 94.1% (87.6% to 97.8%) |
| Ever reported wheezing | 36.1% (25.1% to 48.3%) | 100% (94.8% to 100%) |
| Wheezing reported on two or more occasions | 46.0% (31.8% to 60.7%) | 96.7% (90.7% to 99.3%) |
| Reported wheezing after age 3 | 47.2% (30.4% to 64.5%) | 91.4% (84.4% to 96.0%) |
| Ever reported a frequent wheeze | 50.0% (33.4% to 66.6%) | 93.2% (86.5% to 97.2%) |
| Reported a frequent wheeze after age 3 | 75.0% (50.9% to 91.3%) | 90.9% (84.3% to 95.4%) |
| Combinations of outcomes | | |
| Ever wheezed+ever breathless | 55.6% (38.1% to 72.1%) | 94.3% (88.0% to 97.9%) |
| Ever symptoms+doctor diagnosis | 90.0% (68.3% to 98.8%) | 93.4% (87.4% to 97.1%) |
| Wheezing on ≥2 occasions+doctor diagnosis | 87.0% (89.3% to 98.1%) | 94.9% (89.3% to 98.1%) |
| Ever reported a frequent wheeze+doctor diagnosis | 94.4% (72.7% to 99.9%) | 92.7% (86.6% to 96.6%) |

ALSPAC, Avon Longitudinal Study of Parents and Children; NPV, negative predictive value; PPV, positive predictive value.

**Table 4** Combination of outcomes versus GPRD asthma

| ALSPAC asthma outcome | GP-recorded diagnosis of asthma | |
| --- | --- | --- |
| | No | Yes |
| Ever reported wheezing+ever reported breathlessness | | |
| No | 99 (86.1%) | 6 |
| Yes | 16 | 20 (76.9%) |
| Ever reported symptoms+doctor diagnosis by 7.5 years | | |
| No | 113 (98.3%) | 8 |
| Yes | 2 | 18 (69.2%) |
| Wheezing reported on ≥2 occasions+doctor diagnosis | | |
| No | 112 (97.4%) | 6 |
| Yes | 3 | 20 (76.9%) |
| Ever reported a frequent wheeze+doctor diagnosis | | |
| No | 114 (99.1%) | 9 |
| Yes | 1 | 17 (65.4%) |

ALSPAC, Avon Longitudinal Study of Parents and Children; GP, general practitioner; GPRD, General Practice Research Database.

presented here could be overestimates. However, the relative performance of the measures is unlikely to have been affected.

The main difference between this study and the majority of other asthma validation studies is that the symptom data in the latter were cross-sectional, so wheezing questions generally referred to the past 12 months and coincided with the timing of clinical assessment. These studies have generally concluded that wheezing questions are good at identifying asthmatics. A few studies have looked at 'ever wheezing' to estimate the cumulative incidence of asthma. In these studies, the sensitivity and specificity of 'ever wheezing' ranged from 80.6% to 94.5% and 74.9% to 89.9%, respectively.[5 16–18] However, in each case, this was based on a question asked at only one point in time. Jenkins et al[7] found that reported wheezing was better at identifying adults than children with asthma and suggested that parents unfamiliar with wheezing may misreport other respiratory sounds as wheezing. Asking the question, 'Has your child ever wheezed?' at one time point is more likely to include children with persistent wheezing or asthma diagnosis due to difficulties of parental recall of sporadic wheeze in early childhood.[19] Because we asked questions about wheezing throughout early childhood, this may have included a higher proportion of participants who had viral infections causing wheezing. It has been estimated that 30–50% of preschool children experience episodic viral wheeze at least once.[20] This is supported by a Swedish study, where the prevalence of wheezing ranged from 26.6% among those aged 1–2 to 13.2% among those aged 5–6, whereas the prevalence of doctor-diagnosed asthma at these ages was 3.6% and 5.7%, respectively.[17] Further, results from another study showed that 48.5% of children aged 6 years had ever wheezed but 41% of these had transient early wheezing—at least one lower respiratory tract infection with wheezing before the age of 3 but no wheezing at

6 years; the majority of this group had no increased risk of asthma in later life.[21] Our results regarding reported doctor-diagnosed asthma generally agree with previously published results. One study reported a sensitivity of 76.9% and a specificity of 97.5% for current doctor-diagnosed asthma, as determined from medical records.[17] Two recent studies of the cumulative incidence of childhood asthma determined from GP (UK-based) or health claims data (Canada) compared with parent-reported diagnosis reported high specificities but relatively low sensitivities.[22 23] Finally, de Marco et al[8] found that the question 'Have you ever had asthma?' agreed most closely with clinical diagnosis.

Owing to the lack of firm diagnostic criteria, the measurement of asthma in epidemiological studies is problematic. Using ever wheezing measured longitudinally from early childhood as a proxy for asthma is likely to greatly overestimate its prevalence, whereas reports of a doctor's diagnosis will estimate it reasonably accurately. Our results suggest that there is no 'ideal' way to define asthma in terms of wheezing alone. A measure which correctly identifies the majority of non-asthmatics will not pick up a very large proportion of children with asthma, and vice versa. Having said this, the purpose of an epidemiological study is often to identify risk factors rather than to estimate prevalence. In general, relative risks will be biased towards the null in the presence of non-differential misclassification, and this is particularly dependent on a measure's specificity.[24] Pekkanen and Pearce[3] suggest that a highly specific measure could be achieved by using a combination of BHR and symptoms or BHR, symptoms and reported doctor diagnosis. In our study, combining symptoms or BHR with a doctor's diagnosis certainly achieved high specificities. However, these combinations were not very sensitive and a doctor's diagnosis performed almost as well in terms of specificity but misclassified fewer asthmatics. Further, very high specificities were achieved by combining wheezing outcomes with reports of a doctor's diagnosis, and these also misclassified fewer asthmatics. Peat et al suggest that using reports of a doctor's diagnosis is not a good way to detect participants with current severe asthma. They argue that identifying this group is important for determining risk factors and that this can be achieved with BHR.[25] Our data do not support this, as 65% of those with BHR had never received a diagnosis of asthma. Similarly, only 74% of those with BHR had ever wheezed.

In conclusion, we have found that parental reports of a doctor's diagnosis of asthma agree well with clinical records; we recommend that this question is incorporated into epidemiological questionnaires to supplement symptom data. The use of wheezing to define asthma is more problematic but our results suggest that reasonably high specificity can be achieved by using conditional questions about wheezing, although at the cost of lower sensitivity and failure to identify (probably) milder cases. We have also shown that linking data from

observational studies to electronic patient records can be an effective means of validating parent-reported data, as well as providing a source of outcome data that may otherwise be missing. However, it should be acknowledged that the success of this methodology relies on good coverage of relevant datasets as well as low levels of dissent from participants to link to these.

**Acknowledgements** The authors are extremely grateful to all the families who took part in this study, the midwives for their help in recruiting them and the whole Avon Longitudinal Study of Parents and Children (ALSPAC) team, which includes interviewers, computer and laboratory technicians, clerical workers, research scientists, volunteers, managers, receptionists and nurses.

**Contributors** RPC carried out the statistical analysis and drafted the manuscript. JH contributed to the design of the study, and was involved interpretation of data and drafting of the paper. AWB established the linkage process and contributed to the drafting of the paper. TVS established the linkage and data management process and reviewed the manuscript. RG cleaned, processed and advised on the ALSPAC asthma data and reviewed the manuscript. JM conceived the study. All authors read and approved the final version of the manuscript.

**Funding** This work was supported by the Wellcome Trust (WT086118/Z/08/Z). The UK Medical Research Council (MRC), the Wellcome Trust and the University of Bristol currently provide core funding for ALSPAC (WT092731). Data collection is funded from a wide range of sources.

**Competing interests** JM is the Principal Investigator of the Project to Enhance ALSPAC through Record Linkage (PEARL).

**Ethics approval** Ethical approval was obtained from the ALSPAC Ethics and Law Committee, Local Research Ethics Committees and the National Health Service (NHS) National Information Governance Board (NIGB) Ethics and Confidentiality Committee.

**Provenance and peer review** Not commissioned; externally peer reviewed.

**Data sharing statement** The relevant ALSPAC study data and statistical code are available from RPC; linked General Practice Research Database (GPRD) data are available through the Medicines and Healthcare products Regulatory Agency (MHRA) safe haven as described in the paper. Participants gave informed consent for data sharing. Full information on the ethical and governance stipulations around access to and sharing of ALSPAC data are available on the ALSPAC website.

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
