## [Reviewer comments · BMJ Open]

This paper was submitted to the Thorax and ADC but declined for publication following peer review. The authors addressed the reviewers' comments and submitted the revised paper to BMJ Open. The paper was subsequently accepted for publication at BMJ Open.

ARTICLE DETAILS

TITLE (PROVISIONAL)	Validating childhood asthma in an epidemiological study using linked electronic patient records
AUTHORS	Cornish, Rosie; Henderson, John; Boyd, Andy; Granell, Raquel; Van Staa, Tjeerd; Macleod, John

VERSION 1 – REVIEW

- This manuscript received two reviews at the Thorax but the referees have declined to make their comments public.

VERSION 2 – REVIEW

REVIEWER	Turner, Stephen Department of Child Health, University of Aberdeen
REVIEW RETURNED	02-Aug-2013

GENERAL COMMENTS	The ALSPAC team continue to use their population to extend our current understanding of respiratory disease (and others) and they have to be congratulated for their enthusiasm. I have previously reviewed this manuscript for another journal and see no changes to the text and my previous suggestions also remain unchanged: 1. The novelty of this study is its longitudinal nature but at present it is essentially a cross sectional study. Is it possible to add a longitudinal analysis/survival curve comparing age at which an asthma diagnosis was given by the GP and reported to ALSPAC? Similarly, the authors could do a survival curve for loss of asthma diagnosis in transient cases. I appreciate this may not be possible for all individuals as some may have moved from non GPRD practices but this type of analysis will make best use of the longitudinal data.2. The abstract needs to state that asthma was defined at 7.5 years for ALSPAC but at 9 years from GPRD. Also the abstract needs to state that there were 26 children with asthma in the analysis.3. The authors need to clarify how children with a transient asthma diagnosis in GPRD were handled, eg were they categorised as no asthma at 9. If transient asthma symptoms were categorised as no asthma at 9 in GPRD then I am not sure it is valid to include this group in the analysis of ever symptoms (see comment 1 suggesting a separate group of remittent asthma)4. The authors should clarify why there is this 1.5 year gap. The justification given on page 5 (line 40) that 9 years was chosen to coincide with the timing of ALSPAC (presumably physiological)
--

	measurements seems weak given that asthma diagnoses was ascertained in ALSPAC at 7.5 years. 5. Given that there was a 1.5 year delay between ALSPAC and GPRD time stamps for asthma, how did the authors manage children who gained a GPRD asthma diagnosis between 7.5 and 9 years of age? (were there any?) 6. It would be useful to link FEV1 and skin prick reactivity to GP diagnosed asthma - if only to demonstrate that these are not sensitive or specific. 7. The authors might confirm that they have previously demonstrated that wheeze and asthma at 7 are associated with the usual asthma phenotype, eg reduced lung function, BHR and skin prick reactivity. 8. NHS IC needs to be defined (page4, line 33). I appreciate this is defined in the abstract.
--	--

REVIEWER	Helms, Peter University of Aberdeen, Child Health
REVIEW RETURNED	03-Oct-2013

GENERAL COMMENTS	In view of the potential for linked electronic health records in support of epidemiological and clinical research the current submission is timely. The validity of asthma diagnosis is well chosen as this chronic condition of high prevalence in childhood can be difficult to confirm in epidemiological studies. However In its present form the submission is over-long and its impact is reduced by inclusion of a large number of wheezing outcomes as comparators with doctor diagnosis of asthma from linked primary care records. Although the authors have assumed the primary care record to be the “gold standard” an equally strong case could be made for prospectively gathered data on more than one occasion. Other ways of identifying significant wheezing illness and asthma have been proposed including “treated asthma”. Hence, in this reviewer’s opinion, the title should be something along the lines “level of agreement between childhood asthma reported by parents and in primary care electronic records”. Detailed Comments: Introduction: Page 3, last paragraph: Reference is made to the management of asthma in primary care which implies the use of medicines and hence “treated asthma” which is not dealt with in the present report. Methods: Subjects: It would be helpful for the reader to be supplied with information as to how and when consent by parents and guardians to the linkage of their children’s ALSPAC data to electronic health records was obtained. Linkage between ALSPAC and GPRD: The linkage rate is impressive but it would be helpful for the reader to be given information as to how the linkage was achieved. Was this by deterministic or probabilistic linkage and using what fields ? Results: Whereas the number of subjects in whom doctor diagnosis of asthma could be compared between the two different methods of assessment is of modest size , BHR was only available on a single
--

	occasion in a small number of subjects of whom only 23 appeared to have had “severe BHR” . In view of the known variability of BHR this weakens the case for inclusion of BHR in this analysis. Furthermore, potential bias may have been introduced as individuals agreeing to BHR may have been more likely to have a personal or family history of wheezing or reported asthma. Although BHR may be an informative variable in the long term follow of the ALSPAC cohort it seems out of place here Table 2 contains the core information relating to a doctor diagnosis of asthma and reported wheezing in the past twelve months. The other derived categories add little to the main result and could be removed to a supplementary electronic table and/or outlined briefly in the results with some discussion on the use of various categories of wheezing as outcome measures. Similar comment is relevant to current Table 4. Discussion The concluding paragraph of the Discussion highlights the potential use of routinely acquired health data. However, the use of such data relies on a good coverage of the population of interest within the current CPRD and other relevant datasets such as hospital admissions and dispensed prescriptions. The use of linked data also relies on continuing consent by parents, guardians and subsequently, young people themselves when they reach of age of majority.
--	---

VERSION 2 – AUTHOR RESPONSE

Reviewer: 1

Comments to the Author

The ALSPAC team continue to use their population to extend our current understanding of respiratory disease (and others) and they have to be congratulated for their enthusiasm. I have the following comments:

1. The novelty of this study is its longitudinal nature but at present it is essentially a cross sectional study. Is it possible to add a longitudinal analysis/survival curve comparing age at which an asthma diagnosis was given by the GP and reported to ALSPAC? Similarly, the authors could do a survival curve for loss of asthma diagnosis in transient cases. I appreciate this may not be possible for all individuals as some may have moved from non GPRD practices but this type of analysis will make best use of the longitudinal data.

We do not have information in ALSPAC on when the doctor’s diagnosis was given. We would agree that this question is cross-sectional. However, the symptom data are longitudinal. It would be possible to examine the timing of the diagnosis in GPRD and how this relates to the timing of parent-reported wheezing. However, we feel this goes beyond the scope of this piece of work, which was to examine how different epidemiological definitions of asthma (most of which, but not all, were defined using longitudinal data) performed in terms of their agreement with clinical records. The analysis being suggested is, if I understand it correctly, more about looking at whether agreement between parent-reported measures and clinical diagnosis changes with age. In the paper we mention that there was

only one case where a child had an early diagnosis of asthma but no further evidence in their record. Thus, an analysis of loss of diagnosis is not possible since there would only be this one case.

2. The abstract needs to state that asthma was defined at 7.5 years for ALSPAC but at 9 years from GPRD. Also the abstract needs to clarify that the analysis was performed on 141 children of whom 26 had asthma at 9 years of age.

The latter has been stated; please see below for response regarding this age gap.

3. The authors need to clarify how children with a transient asthma diagnosis in GPRD were handled, e.g. were they categorised as no asthma at 9. If transient asthma symptoms were categorised as no asthma at 9 in GPRD then I am not sure it is valid to include this group in the analysis of ever symptoms (see comment 1 suggesting a separate group of remittent asthma)

This had already been described in the methods section – at the end of the statistical methods.

4. The authors should clarify why there is this 1.5 year gap. The justification given on page 6 (line 33) that 9 years was chosen to coincide with the timing of ALSPAC (presumably physiological) measurements seems weak given that asthma diagnoses was ascertained in ALSPAC at 7.5 years.

Parent-reports of a doctor's diagnosis of asthma were determined at approximately 7.5 years; wheezing and breathlessness were measured up to approximately 103 months (8 years, 7 months); BHR, FEV1 and FVC were measured between the ages of 8 and 9 years. Thus, the timing of GPRD measurement (before age 9) was used to best coincide with these measurements as a whole, rather than the timing of one of these measurements. However, an additional comparison for doctor's diagnosis has now been done, using 8 rather than 9 years as the cut-off to show how much impact this had on the estimated sensitivity and specificity of this measure.

5. Given that there was a 1.5 year delay between ALSPAC and GPRD time stamps for asthma, how did the authors manage children who gained a GPRD asthma diagnosis between 7.5 and 9 years of age? (were there any?).

There were three children who gained a diagnosis while aged 8. Since the exact age at which the questionnaire was completed could not be determined (they were dispatched when the children were aged 7.5 years old but were not necessarily returned straight away), these were simply left as they were (i.e. not analysed separately or omitted). Two of these individuals reported having received a doctor's diagnosis of asthma in the ALSPAC questionnaire; one of them did not.

6. It would be useful to link FEV1 and skin prick reactivity to GP diagnosed asthma - if only to demonstrate that these are not sensitive or specific.

These measures have now been included and results reported as supplementary material.

7. The authors might confirm that they have previously demonstrated that wheeze and asthma at 7 are associated with the usual asthma phenotype, eg reduced lung function, BHR and skin prick reactivity.

I can confirm that they are associated in ALSPAC (Associations of wheezing phenotypes in the first 6 years of life with atopy, lung function and airway responsiveness in mid-childhood; Henderson J et al;

Thorax 2008;63:974-980 doi:10.1136/thx.2007.093187). However, since one of the key issues discussed in the introduction is that such measures are quite variable both between and within children with asthma and cannot be reliably used as diagnostic criteria, this reference does not seem directly relevant to this paper.

8. NHS IC needs to be defined (page 5, line 3). I appreciate this is defined in the abstract.

This has been done.

Reviewer: 2

Comments to the Author

In view of the potential for linked electronic health records in support of epidemiological and clinical research the current submission is timely. The validity of asthma diagnosis is well chosen as this chronic condition of high prevalence in childhood can be difficult to confirm in epidemiological studies. However in its present form the submission is over-long and its impact is reduced by inclusion of a large number of wheezing outcomes as comparators with doctor diagnosis of asthma from linked primary care records.

The paper has been shortened by removing some of the wheezing results and results for BHR to supplementary tables.

Although the authors have assumed the primary care record to be the “gold standard” an equally strong case could be made for prospectively gathered data on more than one occasion. Other ways of identifying significant wheezing illness and asthma have been proposed including “treated asthma”. Hence, in this reviewer’s opinion, the title should be something along the lines “level of agreement between childhood asthma reported by parents and in primary care electronic records”.

We would agree that the primary care record is not a true gold standard and have acknowledged this in the paper. We have decided to stand by our original choice of GP-recorded diagnosis as the standard for the reasons outlined in the paper. In addition, as others have carried out very similar analyses, this makes it easier to compare our results to theirs.

Detailed Comments:

Introduction:

Page 3, last paragraph: Reference is made to the management of asthma in primary care which implies the use of medicines and hence “treated asthma” which is not dealt with in the present report.

We have changed this to say “diagnosed and managed in primary care”. We were not trying to imply treated asthma but were just making the point that any medical information about asthma is likely to appear in an individual’s primary care record.

Methods:

Subjects: It would be helpful for the reader to be supplied with information as to how and when consent by parents and guardians to the linkage of their children’s ALSPAC data to electronic health records was obtained.

Brief details regarding consent has now been added to the relevant subsection of the methods.

Linkage between ALSPAC and GPRD: The linkage rate is impressive but it would be helpful for the reader to be given information as to how the linkage was achieved. Was this by deterministic or probabilistic linkage and using what fields ?

Brief information about the linkage has now been added to the methods section.

Results:

Whereas the number of subjects in whom doctor diagnosis of asthma could be compared between the two different methods of assessment is of modest size, BHR was only available on a single occasion in a small number of subjects of whom only 23 appeared to have had "severe BHR". In view of the known variability of BHR this weakens the case for inclusion of BHR in this analysis. Furthermore, potential bias may have been introduced as individuals agreeing to BHR may have been more likely to have a personal or family history of wheezing or reported asthma. Although BHR may be an informative variable in the long term follow of the ALSPAC cohort it seems out of place here.

Since the first reviewer argued for the inclusion of FEV1 and skin prick data and this reviewer is arguing for the exclusion of BHR, we have compromised by including all of these measures but as supplementary material rather than in the main body of the paper.

Table 2 contains the core information relating to a doctor diagnosis of asthma and reported wheezing in the past twelve months. The other derived categories add little to the main result and could be removed to a supplementary electronic table and/or outlined briefly in the results with some discussion on the use of various categories of wheezing as outcome measures. Similar comment is relevant to current Table 4.

Some of the derived wheezing variables have been removed from Tables 2 and 4 and put into supplementary tables.

Discussion

The concluding paragraph of the Discussion highlights the potential use of routinely acquired health data. However, the use of such data relies on a good coverage of the population of interest within the current CPRD and other relevant datasets such as hospital admissions and dispensed prescriptions. The use of linked data also relies on continuing consent by parents, guardians and subsequently, young people themselves when they reach of age of majority.

We have added a sentence to the discussion to acknowledge these limitations.